# Electrospun Chitosan/Poly (Vinyl Alcohol)/Graphene Oxide Nanofibrous Membrane with Ciprofloxacin Antibiotic Drug for Potential Wound Dressing Application

**DOI:** 10.3390/ijms20184395

**Published:** 2019-09-06

**Authors:** Shuai Yang, Xiaohong Zhang, Dawei Zhang

**Affiliations:** 1Material Science and Engineering College, Northeast Forestry University, Harbin 150040, China; ys930509@163.com; 2School of Medicine, Ningbo University, Ningbo 315211, China; zhangxiaohong1@nbu.edu.cn

**Keywords:** composite nanofibrous membranes, graphene oxide, regulatable drug-release profile, antibacterial activity, cytocompatibility

## Abstract

In this paper, nanofibrous membranes based on chitosan (CS), poly (vinyl alcohol) (PVA) and graphene oxide (GO) composites, loaded with antibiotic drugs, such as Ciprofloxacin (Cip) and Ciprofloxacin hydrochloride (CipHcl) were prepared via the electrospinning technique. The uniform and defect-free CS/PVA nanofibers were obtained and GO nanosheets, shaping spindle and spherical, were partially embedded into nanofibers. Besides, the antibiotic drugs were effectively loaded into the nanofibers and part of which were absorbed into GO nanosheets. Intriguingly, the release of the drug absorbed in GO nanosheets regulated the drug release profile trend, avoiding the “burst” release of drug at the release initial stage, and the addition of GO slightly improved the drug release ratio. Nanofibrous membranes showed the significantly enhanced antibacterial activity against *Escherichia coli*, *Staphylococcus aureus* and *Bacillus subtilis* after the addition of antibiotic drug. Moreover, the drug-loaded nanofibrous membranes exhibited excellent cytocompatibility with Melanoma cells, indicative to the great potential potential for applications in wound dressing.

## 1. Introduction

An ideal wound dressing would provide an optimal healing environment which could act at a maximum healing rate at the occurrence of open wound [1]. Indicative concretely, the wound dressing should possess excellent biomedical properties such as great cytocompatibility, non-toxicity, good antimicrobial activity, sufficient physical protection of the wound site against external intrusions, high porosity to allow for gas exchange and promotion of wound healing [2]. To date, substantial efforts have been made in various production techniques and materials available for the preparation of wound dressings [3]. Electrospun nanofibers, possessing a nanometer scale, are considered as the excellent wound dressing substrates due to the structural similarity to the extracellular matrix, which promote the cell adhesion, migration and proliferation [4]. The conformable and flexible electrospun nanofibrous membranes serve as the effective physical barrier to protect the open wound from further physical damages and contaminations from exogenous microorganisms [5]. Nanofibrous membranes can also create a moist environment around the wound area to promote a wound healing [6]. Meanwhile, the high specific surface area and porous structure of electrospun nanofibers provide high absorptivity, high gas permeation and conformability to contour of the wound area [7]. In addition, due to the high surface area, electrospun nanofibers have a great performance on drug delivery in a controlled and sustained manner [8]. Thus, integration and incorporation of antimicrobial antibiotic, hemostatic drug and therapeutic agent into electrospun nanofibers have been investigated constantly [9,10,11].

In recent years, biomaterial-based wound dressings have been widely used, such as chitin and chitosan (CS), due to the nontoxicity, biocompatibility, biodegradability, antibacterial property, moisture retentionand readily available properties [12,13]. At present, CS nanofibers are increasingly attracting more attention due to the structural advantages conveyed by the nanoscale constituent fibers [14]. Nevertheless, due to the existence of amino groups on the chains, the electrospinnability of CS is extremely poor [15]. Poly (vinyl alcohol) (PVA) has been commonly used to enhance the electrospinnability of CS [16,17,18,19]. Due to the good biocompatibility, hydrophilic properties and biomechanical characteristics [20], CS/PVA electrospun nanofibers have been used widely in biomedical field such as tissue engineering scaffolds, drug delivery and wound dressings [21,22,23,24].

Graphene oxide (GO) has generated great interest in biomedical applications due to its unique physicochemical characteristics and excellent biocompatibility [25,26,27,28]. The presence of epoxide, carbonyl, carboxyl, and hydroxyl groups on GO nanosheets enables facile chemical and biological modification, obtaining GO as a biological substitute in the biomedical field [29,30,31]. Extremely, GO has enhanced antibacterial activity due to the physical damages occurred upon direct contact to bacterial membranes by sharp edges of GO nanosheets [32], which has motivated researchers to carry out regarding the use of GO in an extensive range of biomedical applications including antibacterial materials, bioimaging, biosensors, tissue-engineering scaffoldsand gene/drug delivery [33,34,35].

In this paper, we report a facile method for the preparation of electrospun CS/PVA/GO nanofibrous membranes for potential wound dressing applications, and the enhanced antibacterial activity of nanofibrous membranes was obtained by the addition of Ciprofloxacin (Cip) and Ciprofloxacin hydrochloride (CipHcl). The structures of drug-loaded nanofibrous membranes were characterized via a scanning electron microscopy (SEM), attenuated total refraction Fourier transform infrared spectroscopy (ATR-FTIR) and X-ray diffraction (XRD). The fluorescence effect was investigated via a fluorescence microscope. Swelling degree was evaluated by a corresponding swelling test. Hydrophilic property was measured by a contact angle measurement. Release characteristic of Cip and CipHcl was measured by the drug release test in vitro. Antibacterial activity against Gram-negative *Escherichia coli*, Gram-positive *Staphylococcus aureus* and *Bacillus subtilis* was investigated. In addition, the cytotoxicity was evaluated via cytotoxicity test (MTT assay).

## 2. Results and Discussion

### 2.1. Morphology of Drug-Loaded Nanofibrous Membranes

SEM images of drug-loaded nanofibrous membranes are shown in Figure 1. The electrospinning process parameters were optimized during the process of the adjustment of applied voltage and proportion of CS, PVA, GO and antibiotic drugs. It could be observed that the uniform and smooth drug-loaded nanofibrous membranes were prepared through the electrospinning process [36]. The nanofibers possessed a relatively narrow mean diameter, of 84.6 ± 24.2 nm. It was noteworthy that the diameter of CS/PVA/CipHcl nanofiber was relatively wide, of 136.6 ± 32.9 nm. In the electrospun process, the addition of water-soluble CipHcl increased the viscosity of electrospun precursor solutions. While the supply voltage and electric field were constant, the diameter of nanofibers increased. GO, due to the excellent adsorption ability, absorbed some CipHcl, stabilizing the solutions parameters. As a result, there was no obvious change of diameter of CS/PVA/GO/CipHcl nanofiber. In addition, comparing with Figure 1a vs. Figure 1b, Figure 1c vs. Figure 1d, Figure 1e vs. Figure 1f, the addition of GO decreased slightly the diameter of nanofibers.

### 2.2. FTIR

Figure 2 shows FTIR spectra of pure CS powder, pure PVA nanofibrous membrane, Cip powder, CipHcl powder and drug-loaded nanofibrous membranes. The peak at 3329 cm^−1^ was concerned with the characteristic peak of O–H and N–H bonds, which was relative wide due to the existence of hydrogen bonds. The peak at 2937 cm^−1^ was assigned to C–H bonds stretching vibration and the peak at 1250 cm^−1^ was was concerned with the bend vibration of C–H bonds on PVA chains. In FTIR spectra of pure CS powder, the peak at 1652 cm^−1^ and 1587 cm^−1^ corresponded to the bending frequency of amide I and amide II bonds. After the addition of Cip, the peak of amide I shifted slightly to the low wavenumber, of 1624 cm^−1^. There were not obvious changes on chemical groups after the addition of Cip and CipHcl. It was due to that the characteristic peaks of amino, carbonyl and carboxyl groups on Cip and CipHcl were sheltered by the corresponding groups which existed in composites before. When the addition of GO, the spectra did not show obvious changes. It could be attributed to that the peaks of secondary amino groups and amide groups formed newly, inducing by the reaction between GO and CS, would be covered by the peaks of amino groups and amide groups which have existed in composites before [37].

### 2.3. XRD

Figure 3 shows XRD curves of pure CS powder, pure PVA nanofibrous membrane, Cip powder, CipHcl powder and drug-loaded nanofibrous membranes. In the XRD curve of CipHcl, five characteristic diffraction peaks appeared at 2θ = 6.7°, 2θ = 10.1°, 2θ = 15.1°, 2θ = 23.1° and 2θ = 25.1°. In the XRD curve of Cip, three characteristic diffraction peaks appeared at 2θ = 14.3°, 2θ = 20.6° and 2θ = 25.3°. It could be obtained that CipHcl and Cip both possessed relative regular crystal structures. The diffraction peak of pure CS was at 2θ = 19.8°, and in the XRD curve of PVA, the diffraction peaks occurred at 2θ = 19.2° and 2θ = 22.2°. In Figure 3, after the addition of CS, the diffraction peaks of PVA widened, which was concerned with the effect of hydrogen bonds and the great compatibility between CS and PVA. With the addition of CipHcl and Cip, XRD curves of nanofibrous membranes didn’t change, which could indicate that the addition of CipHcl and Cip would not affect the original crystal structures of nanofibers. The content of drugs was relative less, and the hydrogen bonds could be formed between CipHcl, Cip and CS, PVA, conducive to the well-dispersed of drugs in polymer metrix. Besides, the adsorption of drugs by GO was adverse to the gathering and formation of crystalline structure of drug molecules In addition, the addition of GO destroyed the crystal structures existed in nanofibers before, widening the diffraction peaks of nanofibrous membranes.

### 2.4. Fluorescence Effect

The fluorescence effect of the drug could indicate the location and distribution of Cip and CipHcl in wound area. Fluorescence micrographs of drug-loaded nanofibrous membranes are shown in Figure 4. In Figure 4a, CS/PVA nanofibrous membrane did not show the fluorescence effect. Nevertheless, the membraneshowed the fluorescence effect after the addition of GO, as shown in Figure 4b. It could be attributed to the fluorescence characteristic of GO [38]. In Figure 4c, due to the strong fluorescence characteristic of CipHcl, CS/PVA/CipHcl nanofibrous membranes showed the enhanced fluorescence effect. It is worth noting that when GO and Cip were both loaded into nanofiber, the drug-loaded nanofibrous membranes did not show the fluorescence effect. It might be concerned with the adsorption of CipHcl by GO nanosheets during the process of preparation of electrospun precursor solutions. The fluorescence quenching effect of GO eliminated the fluorescence effect of CipHcl and GO [39]. As a result, the nanofibrous membrane did not show the fluorescence effect, as shown in Figure 4d. In Figure 4e, CS/PVA/Cip nanofibrous membrane showed the fluorescence effect but CS/PVA/GO/Cip nanofibrous membrane did not show the fluorescence effect, as shown in Figure 4f.

### 2.5. Swelling

The swelling effect of drug-loaded nanofibrous membranes could be conducive to the drug release, and the swelling degree is shown in Figure 5. It could be obtained that the swelling degree of CS/PVA nanofibrous membrane reached to the maximum, of 299%. The swelling degree decreased after the addition of GO, due to the crosslinking reaction between amino groups on CS chains and epoxy, carboxyl groups on GO nanosheets. After the addition of CipHcl, the swelling degree of nanofibrous membrane decreased from 299% to 270%. It might be attributed to the fact that the groups on water-soluble CipHcl (amino, carbonyl and carboxyl groups) would interact with the groups on CS and PVA chains. The hydrogen bonds between drugs and the polymer matrix decreased the swelling degree. The addition of Cip further decreased the swelling degree, of 70%, which might be concerned with the distribution of the fillers in nanofibers. The liposoluble Cip mainly distributed on the surface of nanofiber, restricting the adsorption of solvent by the nanofibers. Nevertheless, the distribution of water-soluble CipHcl in the nanofiber was relatively homogeneous. As a result, the swelling degree of CS/PVA/Cip nanofibrous membrane was relatively lower. The swelling degree of CS/PVA/GO/Cip nanofibrous membrane was at the minimum, of 229%.

### 2.6. Contact Angle

Figure 6 shows the hydrophility test results of drug-loaded nanofibrous membranes. It could be obtained that the contact angle of CS/PVA nanofibrous membrane was 49.6°, which decreased when GO was added, to 43.0°. It might be concerned with the distance between nanofibers increased after the addition of GO, water drop showed wetting state on the surface of nanofibrous membrane. When CipHcl and Cip were added, the contact angle increased, to 53.1° and 49.3°, respectively. The contact angle of CS/PVA/Cip nanofibrous membrane was less than it of CS/PVA/CipHcl nanofibrous membrane, which might be attributed to that the distribution of drugs in nanofibers. The liposoluble Cip tended to move to the surface of nanofiber and mainly distributed on the surface, forming the homogeneous coating on the nanofibrous membrane, conducive to the contact between water drop and membrane surface. As a result, the contact angle of CS/PVA/Cip nanofibrous membrane was relatively lower. The hydrophility of CS/PVA/GO/Cip nanofibrous membrane was 45.0°.

### 2.7. In Vitro Drug Release

Figure 7 shows the in vitro drug release curves of drug-loaded nanofibrous membranes. It could be obtained that CS/PVA/Cip nanofibrous membrane and CS/PVA/CipHcl nanofibrous membrane showed a similar drug release profile trend. At the initial state, the drug release ratio was relative fast. At 4 h, the release ratios of CS/PVA/Cip nanofibrous membrane and CS/PVA/CipHcl nanofibrous membrane were 43.6% and 35.2%, respectively. Then, the release ratio slowed down (6 h) and reached a steady stage along with the incubation time (168 h). Over 168 h of release, the release ratios of 91.1% and 59.0% were for CS/PVA/Cip nanofibrous membrane and CS/PVA/CipHcl nanofibrous membrane, respectively. The significant difference of drug release ratios could be attributed to the distribution of Cip and CipHcl in nanofibers. The liposoluble Cip mainly distributed on the surface of nanofiber, promoting the release of Cip. Nevertheless, the distribution of water-soluble CipHcl in the nanofiber was relatively homogeneous. As a result, the release ratio of CS/PVA/Cip nanofibrous membrane was relative higher.

After the addition of GO, the drug release profile trend of the nanofibrous membranes changed. The “burst” release slowed down at the initial stage. At 4 h, the release ratios of CS/PVA/GO/Cip nanofibrous membrane and CS/PVA/GO/CipHcl nanofibrous membrane were 18.7% and 30.0%. Then, the release ratio slowed down at 12 h and reached a steady stage along with the incubation time (168 h). It was worth noting that the release rates of GO-loaded nanofibrous membranes were faster than the corresponding membrane without GO. Over 168 h of release, the release ratios of 96.5% and 62.1% were for CS/PVA/GO/Cip nanofibrous membrane and CS/PVA/GO/CipHcl nanofibrous membrane, which were more than the release ratios of the corresponding membranes without GO. In the preparation process of electrospun precursor solutions, Cip and CipHcl were absorbed into GO nanosheets. At the release initial stage, the drugs absorbed into GO nanosheets would not release into the environment. When the release continued a period time (12 h), the drugs absorbed in GO nanosheets desorbed and released into the environment. Thus, the drug release profile trend of nanofibrous membranes was regulated, avoiding the “burst” release at the initial stage. In addition, the addition of GO increased the distance between nanofibers, promoting the release of the drugs encapsulated into the nanofibers. As a result, the drug release ratio of nanofibrous membranes increased.

### 2.8. Antibacterial Activity

The antibacterial activity of drug-loaded nanofibrous membranes against Escherichia coli (*E. coli*), Staphylococcus aureus (*S. aureus*) and Bacillus subtilis (*B. subtilis*) was investigated through inhibition zone tests, and the results are shown in Figure 8. For *E. coli*, it could be observed, obviously, that the antibacterial activity of CS/PVA nanofibrous membrane was relatively strong, whose radius of the inhibition zone was 9.77 mm. After the addition of GO, the antibacterial activity weakened due to the consumption of amino groups on CS chains which reacted with the epoxy and carboxyl groups on GO nanosheets. When CipHcl was loaded into the nanofiber, the nanofibrous membrane exhibited the better antibacterial activity. The radius of the inhibition zone of CS/PVA/CipHcl nanofibrous membrane and CS/PVA/GO/CipHcl nanofibrous membrane were 4.5 and 6.86 mm, respectively. It was worth noting that the extremely enhanced antibacterial activity was obtained when the addition of Cip. The radius of the inhibition zone of CS/PVA/Cip nanofibrous membrane and CS/PVA/GO/Cip nanofibrous membrane were 19.9 and 19.5 mm, respectively. It might be concerned with the distribution of Cip in nanofibers. The liposoluble Cip mainly distributed on the surface of nanofiber, conducive to the release of Cip, promoting the enhanced antibacterial activity. For *S. aureus*, CS/PVA nanofibrous membrane exhibited the antibacterial activity and its radius of inhibition zone was 2.21 mm. The antibacterial activity of CS/PVA/GO nanofibrous membrane was relative weak, which radius of inhibition zone was 0.27 mm. The radius of inhibition zone of CS/PVA/CipHcl nanofibrous membrane and CS/PVA/GO/CipHcl nanofibrous membrane were 3.2 and 2.56 mm, respectively. Meanwhile, the Cip-loaded nanofibrous membranes showed the enhanced antibacterial activity. The radius of inhibition zone of CS/PVA/Cip nanofibrous membrane and CS/PVA/GO/Cip nanofibrous membrane were 16.2 and 15.3 mm, respectively. For *B. subtilis*, the radius of inhibition zone of CS/PVA nanofibrous membrane was 6.48 mm. After the addition of GO, the radius of inhibition zone was 0.44 mm. The radius of inhibition zone of CS/PVA/CipHcl nanofibrous membrane and CS/PVA/GO/CipHcl nanofibrous membrane were 6.99 and 11.7 mm, respectively. When Cip was added, the antibacterial activity extremely strengthened. The radius of inhibition zone were 19.2 and 18.7 mm, corresponding to CS/PVA/Cip nanofibrous membrane and CS/PVA/GO/Cip nanofibrous membrane, respectively.

It could be obtained that the antibacterial activity of drug-loaded nanofibrous membranes against *E. coli* and *B. subtilis* was stronger, while the antibacterial activity against *S. aureus* was relatively weaker. The prepared drug-loaded nanofibrous membranes have broad-spectrum and excellent antibacterial activity, which could effectively inhibit the growth and reproduction of the bacteria at wound sites, promoting the application in the field of wound dressings.

### 2.9. Cytotoxicity

Melanoma cells (MCs) were used to evaluate the cytotoxicity of drug-loaded nanofibrous membranes, and the test results are shown in Figure 9. It could be obtained that the viability of MCs was 121%, 126% and 122%, after culturing without drug-loaded nanofibrous membranes for 24, 48 and 72 h. When MCs were cultured with the drug-loaded nanofibrous membranes for 24 h, the cell viability of cells treated with CS/PVA nanofibrous membrane, CS/PVA/GO nanofibrous membrane, CS/PVA/CipHcl nanofibrous membrane and CS/PVA/GO/Cip nanofibrous membrane was similar. The cell viability of cells treated with CS/PVA/GO/CipHcl nanofibrous membrane and CS/PVA/Cip nanofibrous membrane was slightly lower, still over 110%. After 48 h, the viability of cells treated with drug-loaded nanofibrous membranes was relatively higher. The minimum was 125% and the maximum was up to 132%. After 72 h, the cell viability was over 120%. The relatively high cell viability indicated that the drug-loaded nanofibrous membranes were relatively nontoxic. The nontoxicity and great cell compatibility adapted the drug-loaded nanofibrous membranes to the widely biomedical wound dressing.

## 3. Experimental

### 3.1. Materials

CS (deacetylation degree 90–91%, average molar mass about 1.6 × 10^5^) was supplied by Zhejiang Golden-Shell Pharmaceutical Co., Ltd. (Yuhuan, China). PVA (polymerization degree 1700, alcoholysis degree 87–89%) was obtained from Aladdin Industrial, Inc (Shanghai, China). Glacial acetic acid was purchased from Tianjin Zhiyuan Reagent Co., Ltd. (Tianjin, China). Natural graphite powder (5000 mesh) was commercially available from Qingdao Tianyuan graphite Co., Ltd (Qingdao, China). H_2_SO_4_ (98 wt %) was purchased from Beijing Chemical Works (Beijing, China). NaNO_3_ was supplied by Tianjin Kermel Co., Ltd. (Tianjin, China). KMnO_4_ was obtained from Tianjin Chemical Reagent Factory (Tianjin, China) and H_2_O_2_ (30%) were obtained from Xilong Chemical Co. Ltd. (Guangzhou, China). Phosphate-buffered saline (PBS, pH 7.2–7.4, 0.01 mol·L^−1^) was purchased from Beijing Leagene Biotech. Co., Ltd. (Beijing, China). The antibiotic drugs, Ciprofloxacin (Cip) and Ciprofloxacin hydrochloride (CipHcl) were obtained from Beijing Pharmacodia Co., Ltd. (Beijing, China). The gram-negative Escherichia coli (ATCC 8739), Gram-positive *S. aureus* (CMCC 26003) and *B. subtilis* (ATCC 6633) were obtained from Guangdong culture collection center (Guangzhou, China). Melanoma cells (A375) were supplied by Shanghai Bioleaf Biotech Co., Ltd. (Shanghai, China). All of the materials, unless otherwise stated, were of analytical grade and all of the materials were used as received without further treatment.

### 3.2. Preparation of GO Dispersion

The GO was prepared from natural graphite powder by a previously reported synthesis method [40]. First, natural graphite powder was dried at 110 °C for 12 h. Graphite powder and NaNO_3_ were placed in the cold (0 °C) H_2_SO_4_ (98 wt %). Then KMnO_4_ was added gradually with stirring, and the temperature of mixture was kept constant, about of 12–14 °C. The reaction went on for 3.5 h and then the mixture was heated to 35–40 °C for an additional 30 min. Distilled water was slowly added to mixture and the temperature was maintained in the range of 45–50 °C. Afterwards, the reaction went on at 90 °C for 30 min and it was terminated by the addition of distilled water and 30% H_2_O_2_ solution. The mixture was filtered and washed with distilled water for several times until the pH value of solution reached 6–7. Afterwards, the mixture was centrifuged at 5000 rpm for 5 min and the solid content of the sediment was determined. GO dispersion (1 wt %) was prepared by the addition of appropriate amount of distilled water and it was treated with ultrasonic which intended to gain a uniform and homogeneous dispersion.

### 3.3. Preparation of Electrospun Precursor Solutions

CS solution was prepared by dissolving 5 wt % of CS powder in distilled water with 5% *v*/*v* acetic acid. PVA powder was dissolved into distilled water at 70 °C, stirring to obtain PVA solution (5 wt %). Cip dispersion (5 wt %) and CipHcl solution (5 wt %) were prepared via dissolving powder in distilled water with an ultrasonic treatment. The mass ratio of CS, PVA and GO was 1:9:0.03, and the content of antibiotic drug (Cip, CipHcl) was 3 wt %. Afterwards, the prepared electrospinning precursor solutions were adequately and slowly stirred, avoiding the formation of bubble in the solution. It was worth noting that the solutions contained Cip should be treated avoiding light.

### 3.4. Preparation of Drug-Loaded Nanofibrous Membranes

The prepared electrospun precursor solutions were transferred into a 10 mL syringe connected to a flat-end metal needle with inner diameter of 0.9 mm. The applied voltage was 23 kV and the distance between needle tip and collector was 15 cm. The needle, displayed with a horizontal plane of 15°, was perpendicular to the collecting plate. Thus the electrospinning process could be carried out by the pushing from gravity of the electrospun precursor solution. The temperature was 25 °C and relative humidity was less than 45%. The electrospinning process continued for 12 h and the prepared nanofibrous membranes were dried overnight at room temperature.

### 3.5. Characterization

The morphology of prepared nanofibrous membranes was observed by field emission scanning electron microscopy (JSM-7500F, Jeol, Tokyo, Japan). Prior to the analysis, the samples were sputter-coated with gold for better conductivity during imaging. The mean diameters of prepared nanofibers were determined by measuring 100 different fibers randomly selected using Digimizer software (V4.5.1, Belgium, Germany). The chemical structures of prepared nanofibrous membranes were investigated by attenuated total refraction Fourier transform infrared spectroscopy (Tensor Ⅱ FTIR, Bruker). The spectral range was 4000–400 cm^−1^ with a resolution of 4 cm^−1^. XRD analysis was used to determine the crystallinity of prepared nanofibrous membranes. XRD patterns were obtained on a XRD analyzer (D/max-2200VPC, Rigaku, Tokyo, Japan) and a scanning rate was of 5° min^−1^ over a 2θ range of 5–55°. The fluorescence effect of prepared nanofibrous membranes was investigated via a fluorescence microscope (Axioskop 2 plus, Zeiss, Oberkochen, Germany).

### 3.6. Swelling Measurement

The swelling degree of the prepared nanofibrous membranes was measured via a corresponding swelling test. The samples were dried at 40 °C for 24 h in a vacuum oven (−0.1 MPa) and the mass (*M_d_*) of samples was measured. The test was carried out in the release medium (PBS, pH 7.2–7.4) at the room temperature for 72 h. The swollen mass (*M*_s_) of nanofibrous mats was determined. The swelling degree of nanofibrous membranes was calculated by Equation (1).
(1)$$\left(\frac{M_{s}-M_{d}}{M_{d}}\right)\times 100\%=Degree\;of\;swelling\;(\%)$$
where *M_d_* was the mass of dried nanofibrous membranes before swelling. *M_s_* was the mass of swollen nanofibrous membranes which was wiped dry with filter paper.

### 3.7. Contact Angle Measurement

The hydrophilicity of the prepared nanofibrous membranes was investigated via a contact angle system (OCA 20, Dataphysics, Beijing, China). A drop of water (5 μL) was deposited on the membrane surface and the drop shape was recorded by a digital camera (OCA 20, Dataphysics, Beijing, China).

### 3.8. In Vitro Drug Release Assay

The release characteristic in vitro of Cip and CipHcl from prepared nanofibrous membranes was characterized at 37 °C in release medium (PBS, pH 7.2–7.4). The known mass (8 mg) of nanofibrous membranes was immersed in 15 mL PBS with stirring constantly. At designated time intervals, 2.0 mL solution was taken out, and an equal amount of fresh buffer solution was supplemented, assuring the amount of release medium constant during the release process. The amount of Cip and CipHcl which was released from nanofibrous membranes into PBS solution was quantified by capturing UV-Vis absorption spectra (TU-1901 Persee, Wuxi, China) at a wavelength of 280 nm. The release experiments of each sample were performed in triplicate, and the mean values were reported.

### 3.9. Antibacterial Activity Assay

The antibacterial activity of the prepared nanofibrous membranes was investigated against *E. coli* (ATCC 8739), *S. aureus* (CMCC 26003) and *B. subtilis* (ATCC 6633) via agar disk diffusion method. The antibacterial activity was reflected from the radius of inhibition zones (mm). The plates, filled with sterilized Luria–Bertani agar medium, were inoculated with microbial strains and the samples were cut into small rectangle pieces (10 × 10 mm^2^). Afterwards, the samples were put on the inoculated agar and the plates were incubated at 37 °C for 12 h in an incubator. After incubation, the radius of inhibition zones formed was measured via Digimizer software.

### 3.10. Cytotoxicity Assay

Melanoma cells were used to evaluate the cytotoxicity of prepared nanofibrous membranes through MTT assay. A certain quality (8 mg) nanofibrous membranes were cut up and UV-sterilized. The treated membranes were immersed into the serum-containing medium at 37 °C for 12 h. The cells were seed into 96-plate and incubated in humid environment (5% CO_2_, 95% air, 37 °C) for 24 h. Afterwards, the cells, treated with the sample supernatants, were used as experiments. While the cells which were not treat with the supernatants were used as controls. The plates were cultured for 24, 48 and 72 h and the cell viability was tested by MTT assays.

## 4. Conclusions

In this paper, continuous uniform CS/PVA/GO nanofibrous membranes loaded with CipHCl or Cip were prepared via electrospinning. The structures of nanofibrous membranes were characterized by SEM, FTIR and XRD, and the mean diameter of nanofibers was 84.6 ± 24.2 nm. GO, CipHcl and Cip possessed the fluorescence effect which could be detected via the fluorescence microscope. Nevertheless, when the GO and antibiotic drugs were both loaded into nanofibers, nanofibrous membranes did not show the fluorescence effect, which might be concerned with the fluorescence quenching effect of GO. In a swelling test, the addition of GO, CipHcl and Cip decreased the swelling degree of nanofibrous membranes. In contact angle test, nanofibrous membranes showed the good hydrophilicity. After the addition of CipHcl and Cip, the contact angle increased slightly. Drug-loaded nanofibrous membranes exhibited the excellent drug release ratio and drug release rate. The addition of GO regulated the drug release profile trend of nanofibrous membranes, avoiding the “burst” release at initial stage. In addition, the addition of GO increased the distance between nanofibers, promoting the release of the drugs encapsulated into the nanofibers, increasing the drug release ratio. The addition of antibiotic drugs significantly enhanced the antibacterial activity of nanofibrous membranes, and the nanofibrous membranes were observed to achieve desirable antibacterial activity against *E. coli*, *S. aureus* and *B. subtilis*. In the cytotoxicity test, the cell viability of all the samples was over 110%, indicating that the drug-loaded nanofibrous membranes did not affect the viability of MCs, showing the excellent cytocompatibility. The entire results suggested that the prepared drug-loaded CS/PVA/GO nanofibrous membranes were great potential candidates as wound dressings.

## Figures and Tables

**Figure 1 ijms-20-04395-f001:**
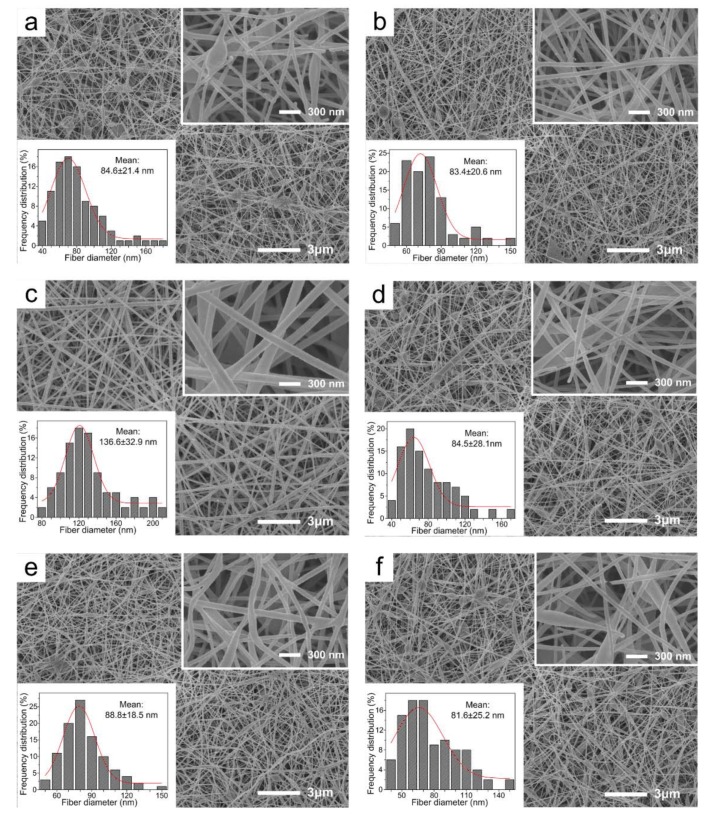
SEM images of drug-loaded nanofibrous membranes: (**a**) chitosan (CS)/ poly (vinyl alcohol) (PVA) nanofibrous membrane, (**b**) CS/PVA/graphene oxide (GO) nanofibrous membrane, (**c**) CS/PVA/Ciprofloxacin hydrochloride (CipHcl) nanofibrous membrane, (**d**) CS/PVA/GO/CipHcl nanofibrous membrane, (**e**) CS/PVA/ Ciprofloxacin (Cip) nanofibrous membrane, (**f**) CS/PVA/GO/Cip nanofibrous membrane.

**Figure 2 ijms-20-04395-f002:**
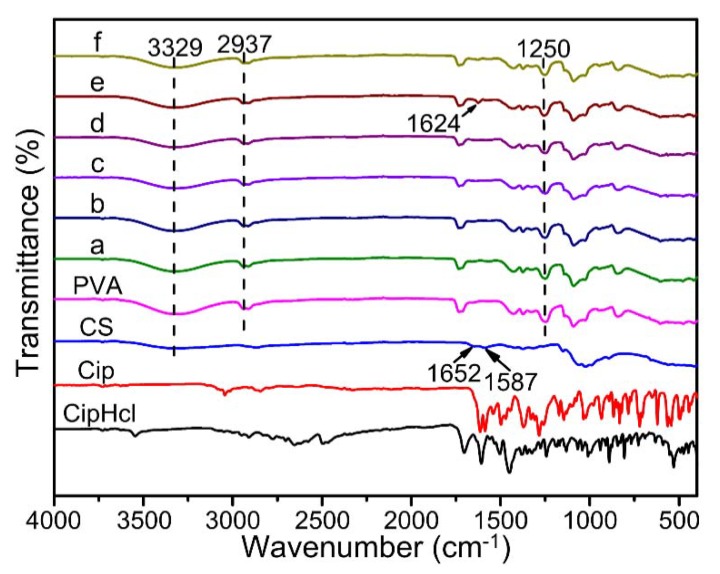
Fourier transform infrared spectroscopy (FTIR) spectra of pure CS powder, pure PVA nanofibrous membrane, Cip powder, CipHcl powder and drug-loaded nanofibrous membranes: (**a**) CS/PVA nanofibrous membrane, (**b**) CS/PVA/GO nanofibrous membrane, (**c**) CS/PVA/CipHcl nanofibrous membrane, (**d**) CS/PVA/GO/CipHcl nanofibrous membrane, (**e**) CS/PVA/Cip nanofibrous membrane, (**f**) CS/PVA/GO/Cip nanofibrous membrane.

**Figure 3 ijms-20-04395-f003:**
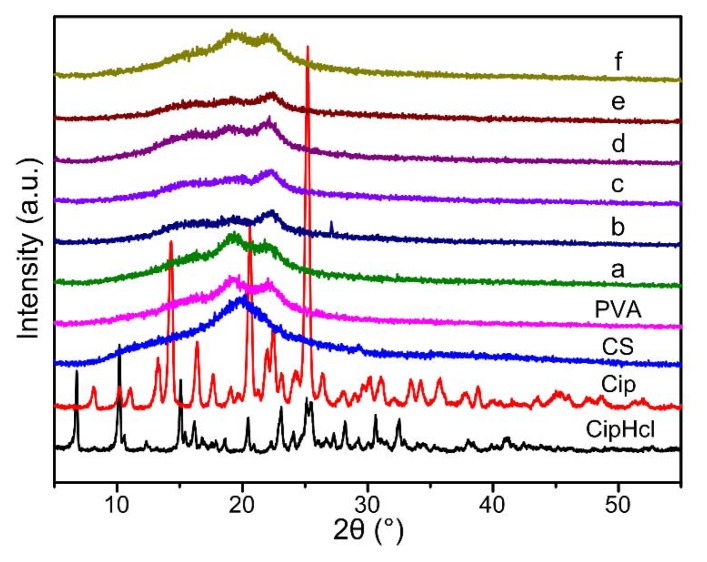
X-ray diffraction (XRD) diffraction patterns of pure CS powder, pure PVA nanofibrous membrane, Cip powder, CipHcl powder and drug-loaded nanofibrous membranes: (**a**) CS/PVA nanofibrous membrane, (**b**) CS/PVA/GO nanofibrous membrane, (**c**) CS/PVA/CipHcl nanofibrous membrane, (**d**) CS/PVA/GO/CipHcl nanofibrous membrane, (**e**) CS/PVA/Cip nanofibrous membrane, (**f**) CS/PVA/GO/Cip nanofibrous membrane.

**Figure 4 ijms-20-04395-f004:**
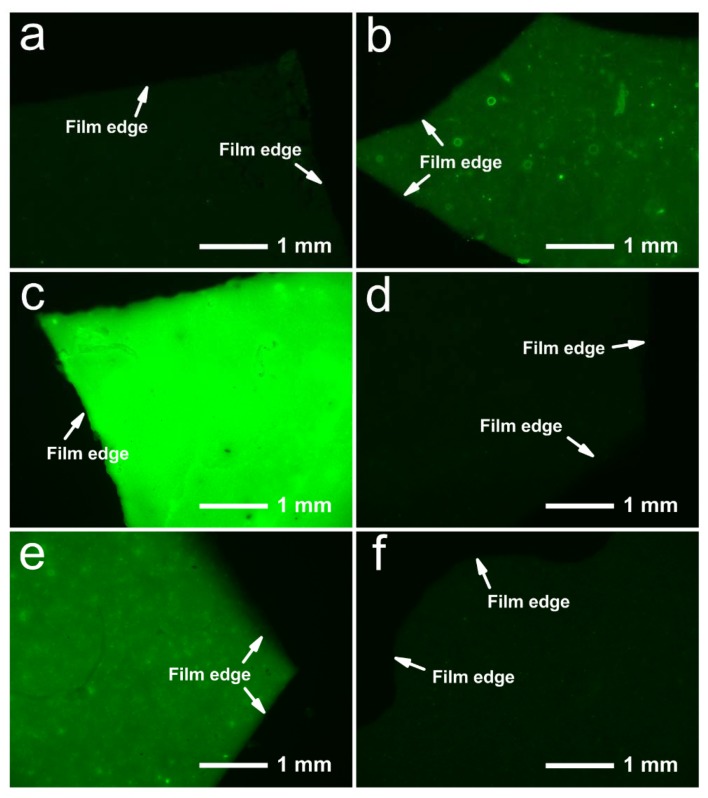
Fluorescence micrographs of drug-loaded nanofibrous membranes: (**a**) CS/PVA nanofibrous membrane, (**b**) CS/PVA/GO nanofibrous membrane, (**c**) CS/PVA/CipHcl nanofibrous membrane, (**d**) CS/PVA/GO/CipHcl nanofibrous membrane, (**e**) CS/PVA/Cip nanofibrous membrane, (**f**) CS/PVA/GO/Cip nanofibrous membrane.

**Figure 5 ijms-20-04395-f005:**
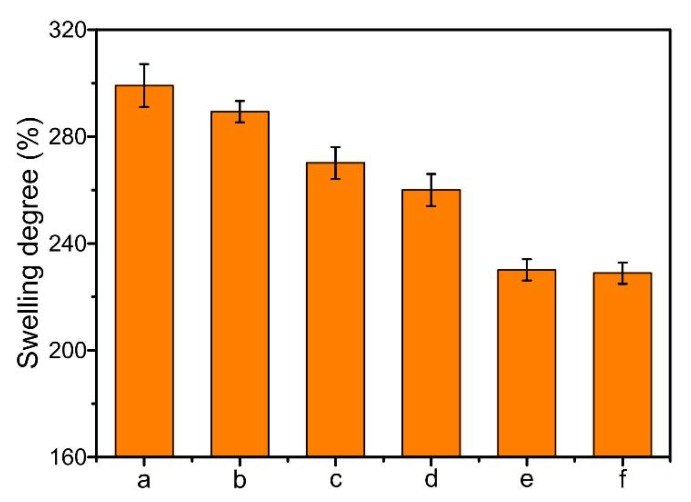
Swelling degree of drug-loaded nanofibrous membranes: (**a**) CS/PVA nanofibrous membrane, (**b**) CS/PVA/GO nanofibrous membrane, (**c**) CS/PVA/CipHcl nanofibrous membrane, (**d**) CS/PVA/GO/CipHcl nanofibrous membrane, (**e**) CS/PVA/Cip nanofibrous membrane, (**f**) CS/PVA/GO/Cip nanofibrous membrane.

**Figure 6 ijms-20-04395-f006:**
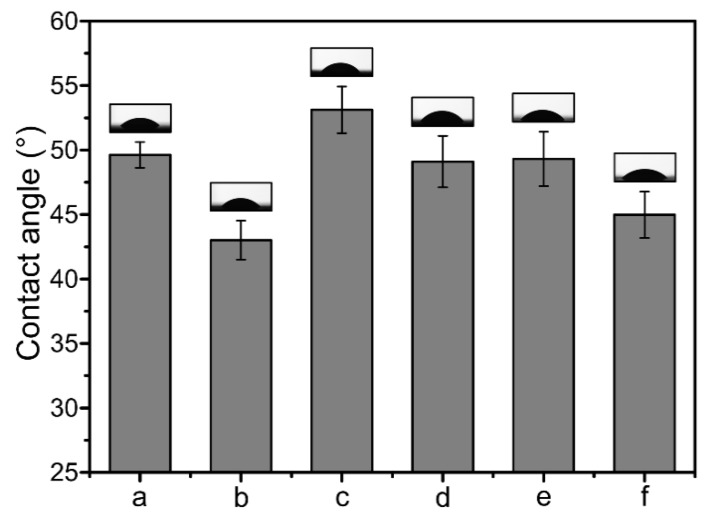
Contact angle values and bars of drug-loaded nanofibrous membranes: (**a**) CS/PVA nanofibrous membrane, (**b**) CS/PVA/GO nanofibrous membrane, (**c**) CS/PVA/CipHcl nanofibrous membrane, (**d**) CS/PVA/GO/CipHcl nanofibrous membrane, (**e**) CS/PVA/Cip nanofibrous membrane, (**f**) CS/PVA/GO/Cip nanofibrous membrane. Inset: water drop images for each membrane surface.

**Figure 7 ijms-20-04395-f007:**
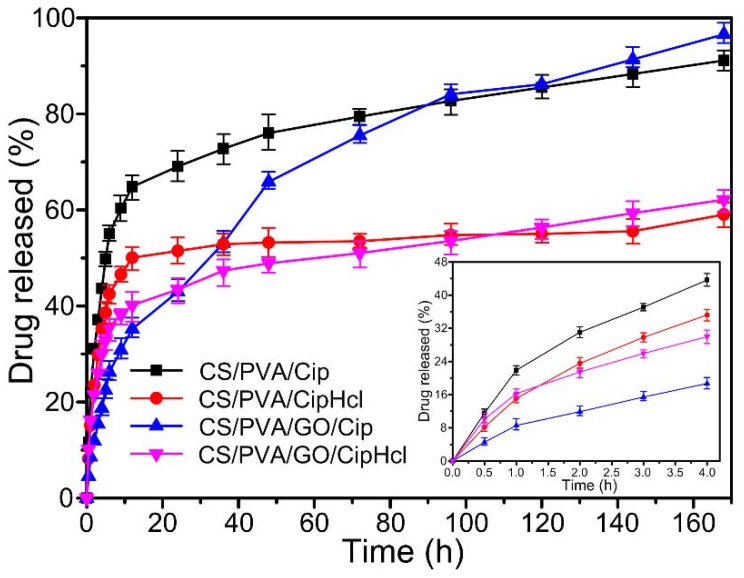
In vitro drug release curves of drug-loaded nanofibrous membranes.

**Figure 8 ijms-20-04395-f008:**
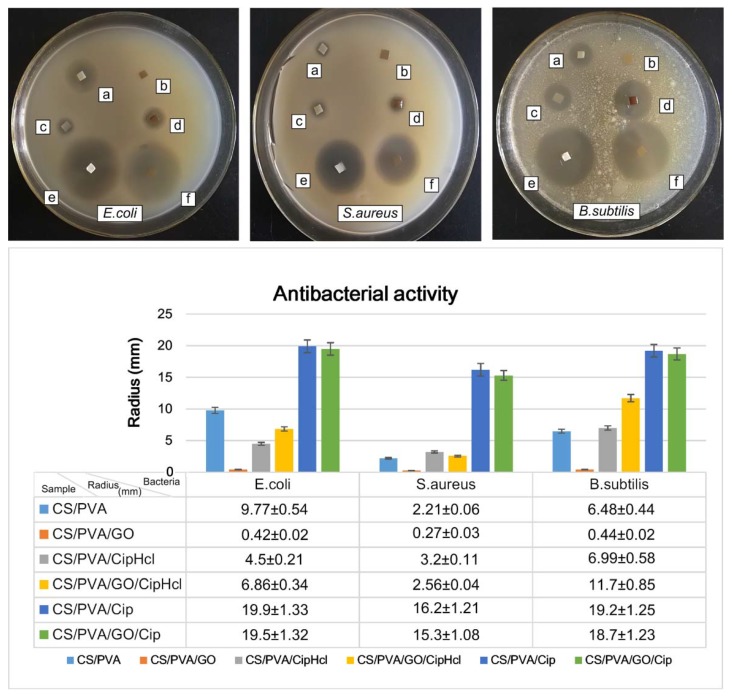
Photographic images and radius statistical diagram of inhibition zones of drug-loaded nanofibrous membranes: (**a**) CS/PVA nanofibrous membrane, (**b**) CS/PVA/GO nanofibrous membrane, (**c**) CS/PVA/CipHcl nanofibrous membrane, (**d**) CS/PVA/GO/CipHcl nanofibrous membrane, (**e**) CS/PVA/Cip nanofibrous membrane, (**f**) CS/PVA/GO/Cip nanofibrous membrane.

**Figure 9 ijms-20-04395-f009:**
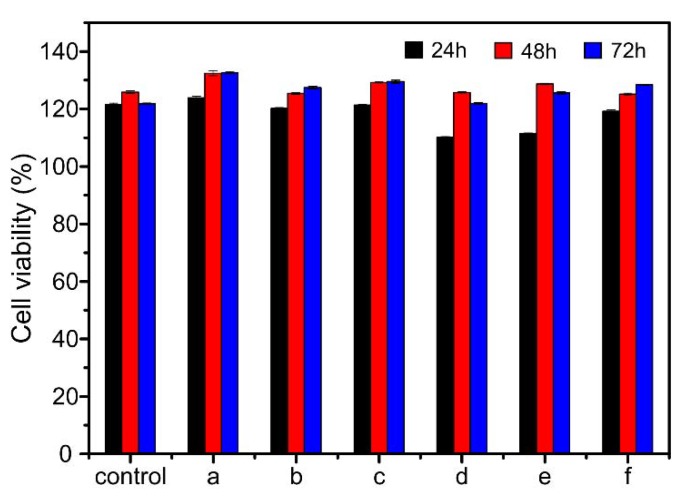
Viability of melanoma cells (MCs) incubated without and with drug-loaded nanofibrous membranes: (**a**) CS/PVA nanofibrous membrane, (**b**) CS/PVA/GO nanofibrous membrane, (**c**) CS/PVA/CipHcl nanofibrous membrane, (**d**) CS/PVA/GO/CipHcl nanofibrous membrane, (**e**) CS/PVA/Cip nanofibrous membrane, (**f**) CS/PVA/GO/Cip nanofibrous membrane (*p** < 0.05).

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
