# Peer review of "Electrospun Chitosan/Poly (Vinyl Alcohol)/Graphene Oxide Nanofibrous Membrane with Ciprofloxacin Antibiotic Drug for Potential Wound Dressing Application"

_ijms, 2019, doi:10.3390/ijms20184395_

Round 1

Reviewer 1 Report

The reviewed manuscript entitled: “Electrospun Chitosan/Poly (Vinyl Alcohol)/Graphene Oxide Nanofibrous Membrane with Ciprofloxacin Antibiotic Drug for Wound Dressing Application” could be regarded as original research effort focused on preparation of antibiotic loaded antimicrobial composite fibrous mats by electrospinning.

Though the idea is unambitiously not new to the research community its implementation as described evoke some serious concerns and critics, namely:

Introduction part:

The state of the art for pristine chitosan/PVA blend electrospun nfs – there is lack of citation of important pioneering works as for example: Procedia Engineering, Volume 8, 2011, Pages 101-105, among others which results in terms of obtained fibers morphology (defects formation) are contractionary with the relatively uniform fibers presented in the current communication and this need further explanation in the Results and discussion part.

There is also important lack of citation of the most important studies and attempts for stabilization of the obtained water-soluble mats by chemical/physical crosslinking methods both in situ and ex situ and the chemical agents used and proof methodologies for the efficiency of the cross-linking (e.g. SEM observation of the swollen mats after prolonged immersion in water (even worm one), sol-gel fractions, FT-IR proof etc.) which is missing in this research.

“The presence of epoxide, carbonyl, and hydroxyl groups on GO nanosheets enables facile chemical and biological modification, obtaining GO as a biological substitute in the biomedical field [29-31].” Surface available carboxylic acid groups seems to be quite important too…

“Extremely, GO has enhanced antibacterial activity due to the physical damages occurred upon direct contact to bacterial membranes by sharp edges of GO nanosheets [32], which has motivated researchers to carry out regarding the use of GO in an extensive range of biomedical applications including antibacterial materials, bioimaging, biosensors, tissue-engineering scaffolds, and gene/drug delivery [33-35].” This part sound quite disturbing as inorganic anisotropic nanomaterials with sharp edges morphology are notoriously known for their in vivo/ in vitro physical acting biological in particular cancerogenic effect best known example is asbestos but there are also doubts for carbon nanotubes etc.  In the cite paper [32] seems the enhanced antimicrobial effect of GO nanosheets is explained by the catalytic decomposition of H2O2 and the enhanced OH radicals formed.

Experimental:

There important data for the chemical crosslinking of the composite fibers by GO is missing and the vacuum treatment at 40C seems unsatisfactory to generate substantial degree of possible intermolecular chitosan and mostly PVA cross-linking ester/amide (unlikely?) bond formation. Also 0.03 wt.?% GO also seems to low concentration for efficient crosslinking of highly swellable polymers like chitosan and PVA. There is no blank experiment for pristine chitosan/PVA and pure PVA mats, vacuum treated and water immersed.

Cytotoxicity assay test: Here the use of melanoma cell line for in vitro cytotoxicity seems as contractional too as the neoplastic cells are quite different from normal tissue cell lines – environment robust, adaptive, crazy genome etc. Their use in cytotoxicity assay is something like eukaryotic analogy for the use of drug-resistant bacterial strains in standard antimicrobial test…

Results and discussion: Among the aforementioned critics, the instrumental analysis are poorly presented as for ex. the FT-IR and the XRD at such low antibiotic and GO concentrations seems inadequate as the instrumental detection limits etc.

The swelling of the “crosslinked” mats is explained by only chitosan (the minor component) chemical crosslinking by ring opening of the hypothetical surface epoxy GO groups (most of which could nucleophilically interact with the water molecules during the ultrasound purification and aq storage dispersion?). This part is quite doubtful anyway and needs further deep investigation and other type of experimental proof implementation!

Author Response

The reviewed manuscript entitled: “Electrospun Chitosan/Poly (Vinyl Alcohol)/Graphene Oxide Nanofibrous Membrane with Ciprofloxacin Antibiotic Drug for Wound Dressing Application” could be regarded as original research effort focused on preparation of antibiotic loaded antimicrobial composite fibrous mats by electrospinning.

Though the idea is unambitiously not new to the research community its implementation as described evoke some serious concerns and critics, namely:

Introduction part:

The state of the art for pristine chitosan/PVA blend electrospun nfs – there is lack of citation of important pioneering works as for example: Procedia Engineering, Volume 8, 2011, Pages 101-105, among others which results in terms of obtained fibers morphology (defects formation) are contractionary with the relatively uniform fibers presented in the current communication and this need further explanation in the Results and discussion part.

There is also important lack of citation of the most important studies and attempts for stabilization of the obtained water-soluble mats by chemical/physical crosslinking methods both in situ and ex situ and the chemical agents used and proof methodologies for the efficiency of the cross-linking (e.g. SEM observation of the swollen mats after prolonged immersion in water (even worm one), sol-gel fractions, FT-IR proof etc.) which is missing in this research.

“The presence of epoxide, carbonyl, and hydroxyl groups on GO nanosheets enables facile chemical and biological modification, obtaining GO as a biological substitute in the biomedical field [29-31].” Surface available carboxylic acid groups seems to be quite important too…

“Extremely, GO has enhanced antibacterial activity due to the physical damages occurred upon direct contact to bacterial membranes by sharp edges of GO nanosheets [32], which has motivated researchers to carry out regarding the use of GO in an extensive range of biomedical applications including antibacterial materials, bioimaging, biosensors, tissue-engineering scaffolds, and gene/drug delivery [33-35].” This part sound quite disturbing as inorganic anisotropic nanomaterials with sharp edges morphology are notoriously known for their in vivo/ in vitro physical acting biological in particular cancerogenic effect best known example is asbestos but there are also doubts for carbon nanotubes etc.  In the cite paper [32] seems the enhanced antimicrobial effect of GO nanosheets is explained by the catalytic decomposition of H2O2 and the enhanced OH radicals formed.

Experimental:

There important data for the chemical crosslinking of the composite fibers by GO is missing and the vacuum treatment at 40C seems unsatisfactory to generate substantial degree of possible intermolecular chitosan and mostly PVA cross-linking ester/amide (unlikely?) bond formation. Also 0.03 wt.?% GO also seems to low concentration for efficient crosslinking of highly swellable polymers like chitosan and PVA. There is no blank experiment for pristine chitosan/PVA and pure PVA mats, vacuum treated and water immersed.

Cytotoxicity assay test: Here the use of melanoma cell line for in vitro cytotoxicity seems as contractional too as the neoplastic cells are quite different from normal tissue cell lines – environment robust, adaptive, crazy genome etc. Their use in cytotoxicity assay is something like eukaryotic analogy for the use of drug-resistant bacterial strains in standard antimicrobial test…

Results and discussion: Among the aforementioned critics, the instrumental analysis are poorly presented as for ex. the FT-IR and the XRD at such low antibiotic and GO concentrations seems inadequate as the instrumental detection limits etc.

The swelling of the “crosslinked” mats is explained by only chitosan (the minor component) chemical crosslinking by ring opening of the hypothetical surface epoxy GO groups (most of which could nucleophilically interact with the water molecules during the ultrasound purification and aq storage dispersion?). This part is quite doubtful anyway and needs further deep investigation and other type of experimental proof implementation!

Q 1) The state of the art for pristine chitosan/PVA blend electrospun nfs – there is lack of citation of important pioneering works as for example: Procedia Engineering, Volume 8, 2011, Pages 101-105, among others which results in terms of obtained fibers morphology (defects formation) are contractionary with the relatively uniform fibers presented in the current communication and this need further explanation in the Results and discussion part.

Thanks for your comment!

Answer: The paper (Procedia Engineering, Volume 8, 2011, Pages 101-105) was added to citation. We compared this paper with the cited paper. In this paper, the uniform and smooth drug-loaded nanofibrous membranes were prepared due to that the electrospinning process parameters were optimized before through the adjustment of applied voltage and proportion of CS, PVA, GO and antibiotic drugs. The further explanation was added into the Results and discussion part, shown as follows:

The electrospinning process parameters were optimized before through the adjustment of applied voltage and proportion of CS, PVA, GO and antibiotic drugs. It could be observed that the uniform and smooth drug-loaded nanofibrous membranes were prepared through the electrospinning process [37].

Paipitak, K.; Pornpra, T.; Mongkontalang, P.; Techitdheera, W.; Pecharapa, W. Characterization of PVA-Chitosan Nanofibers Prepared by Electrospinning. Procedia Engineering 2011, 8, 101-105.

Q 2) There is also important lack of citation of the most important studies and attempts for stabilization of the obtained water-soluble mats by chemical/physical crosslinking methods both in situ and ex situ and the chemical agents used and proof methodologies for the efficiency of the cross-linking (e.g. SEM observation of the swollen mats after prolonged immersion in water (even worm one), sol-gel fractions, FT-IR proof etc.) which is missing in this research.

Thanks for your comment!

Answer: The SEM images of the swollen nanofibrous membranes which the reviewer commented could be provided. Nevertheless, due to the paper was emphasized on the drug release and antibacterial activity, they were not shown in paper. The weight loss and SEM images of swollen nanofibrous membranes were shown as follows. The weight of nanofibers decreased with the prolongation of immersion time. After 48 hours of immersion, the residual weight of GO loaded nanofibrous membrane was higher. After immersion in PBS buffer, the nanofibers were broken by swelling, and many small fragments were formed in the solution.

Figure 1 Residual weight of the prepared nanofibers after soaking in PBS for designated time (6h, 12h, 24h, and 48h): (a) CS/PVA nanofibrous membrane, (b) CS/PVA/GO nanofibrous membrane, (c) CS/PVA/CipHcl nanofibrous membrane, (d) CS/PVA/GO/CipHcl nanofibrous membrane, (e) CS/PVA/Cip nanofibrous membrane, (f) CS/PVA/GO/Cip nanofibrous membrane.

Figure 2 The morphology of nanofibrous membranes after soaking in PBS for 48h: (a) CS/PVA nanofibrous membrane, (b) CS/PVA/GO nanofibrous membrane, (c) CS/PVA/CipHcl nanofibrous membrane, (d) CS/PVA/GO/CipHcl nanofibrous membrane, (e) CS/PVA/Cip nanofibrous membrane, (f) CS/PVA/GO/Cip nanofibrous membrane.

Q 3) “The presence of epoxide, carbonyl, and hydroxyl groups on GO nanosheets enables facile chemical and biological modification, obtaining GO as a biological substitute in the biomedical field [29-31].” Surface available carboxylic acid groups seems to be quite important too…

Thanks for your comment!

Answer: The carboxyl groups on the edge of GO are indeed important, especially which can react with CS matrix by amidation. According to the reviewer's comment, carboxyl groups were added in this sentence, and the revised sentences are as follows:

The presence of epoxide, carbonyl, carboxyl, and hydroxyl groups on GO nanosheets enables facile chemical and biological modification, obtaining GO as a biological substitute in the biomedical field [29-31].

Q 4) “Extremely, GO has enhanced antibacterial activity due to the physical damages occurred upon direct contact to bacterial membranes by sharp edges of GO nanosheets [32], which has motivated researchers to carry out regarding the use of GO in an extensive range of biomedical applications including antibacterial materials, bioimaging, biosensors, tissue-engineering scaffolds, and gene/drug delivery [33-35].” This part sound quite disturbing as inorganic anisotropic nanomaterials with sharp edges morphology are notoriously known for their in vivo/ in vitro physical acting biological in particular cancerogenic effect best known example is asbestos but there are also doubts for carbon nanotubes etc. In the cite paper [32] seems the enhanced antimicrobial effect of GO nanosheets is explained by the catalytic decomposition of H2O2 and the enhanced OH radicals formed.

Thanks for your comment!

Answer: The reference [32] mainly investigated “graphene quantum dots-band-aids used for wound disinfection”. In whose introduction, the damage induced by nanomaterials was mentioned, reflected by “Graphene and its drivatives have high antibacterial activity due to physical damages occurred upon direct contact to bacterial membranes by sharp edges of graphene sheets; however, the improved oxidative stress induced by graphene-based material can lead to the apoptosis of mammalian cells.” The cited paper was just aimed to convince the antibacterial activity of nanomaterials.

Q 5) There important data for the chemical crosslinking of the composite fibers by GO is missing and the vacuum treatment at 40C seems unsatisfactory to generate substantial degree of possible intermolecular chitosan and mostly PVA cross-linking ester/amide (unlikely?) bond formation. Also 0.03 wt.?% GO also seems to low concentration for efficient crosslinking of highly swellable polymers like chitosan and PVA. There is no blank experiment for pristine chitosan/PVA and pure PVA mats, vacuum treated and water immersed.

Thanks for your comment!

Answer: The crosslinking effect could not react between CS and PVA during the electrospinning process. In composite membranes, the crosslinking reaction occurred between GO and CS, and the content of GO was 3 wt% of CS. The papers which investigated the crosslinking effect between GO and CS or CS/PVA nanofibers were reported previously. The crosslinking effect between GO and CS/PVA nanofibers could be cited by Advanced composites and hybrid materials, 2018, 1(4): 722-730, and crosslinking effect between GO and CS could cited by Applied surface science, 2013, 28: 989-992.

Q 6) Cytotoxicity assay test: Here the use of melanoma cell line for in vitro cytotoxicity seems as contractional too as the neoplastic cells are quite different from normal tissue cell lines – environment robust, adaptive, crazy genome etc. Their use in cytotoxicity assay is something like eukaryotic analogy for the use of drug-resistant bacterial strains in standard antimicrobial test.

Thanks for your comment!

Answer: The comment of reviewer was reasonable, primary fibroblasts and keratinocytes could be used in the experiments. We mainly considered the cytotoxicity of prepared nanofibers, Melanoma cells are a kind of human cells. Once nanofibers have great compatibility to melanoma cells, their cytocompatibility is relatively good.

Q 7) Results and discussion: Among the aforementioned critics, the instrumental analysis are poorly presented as for ex. the FT-IR and the XRD at such low antibiotic and GO concentrations seems inadequate as the instrumental detection limits etc.

Thanks for your comment!

Answer: The comment of reviewer was reasonable, due to the paper was emphasized on the drug release and antibacterial activity, the more accurate instrumental analysis was not necessary in the paper. FT-IR and XRD characterizations were just to characterize the chemical groups and crystalline structure of prepared nanofibers.

Q 8) The swelling of the “crosslinked” mats is explained by only chitosan (the minor component) chemical crosslinking by ring opening of the hypothetical surface epoxy GO groups (most of which could nucleophilically interact with the water molecules during the ultrasound purification and aq storage dispersion?). This part is quite doubtful anyway and needs further deep investigation and other type of experimental proof implementation!

Thanks for your comment!

Answer: The crosslinking effect between GO and CS/PVA nanofibers could be cited by Advanced composites and hybrid materials, 2018, 1(4): 722-730. [38] The corresponding content is as follows:

FTIR spectra of CS powder, pure PVA nanofiber, electrospun CS/PVA nanofiber and electrospun CS/PVA nanofiber with 2.5 wt % GO are shown in Fig.3. The peak at 3640-2978 cm-1 could be attributed to the vibrational stretching of O-H [20] and N-H [21] bonds which became broad due to the existence of hydrogen bonds between CS and PVA. From the FTIR spectrum of CS, the peak at 2862 cm-1 was assigned to C-H stretching [22]. Feature peaks of amide I & amide II groups were observed at 1651 cm-1 and 1587 cm-1 [23], respectively. In the FTIR spectrum of PVA, the peak at 1725 cm-1 was concerned with C=O bond of residual ester groups, and the vibrational bending of C-H bond of PVA was observed at 1252 cm-1 [24]. The peak at 2918 cm-1 was assigned to C-H stretching, which shifted to a slightly lower wavenumber in CS. Comparing with the spectra of CS/PVA nanofiber with and without GO, there was no obvious difference between two curves. In fact, the epoxy groups on GO molecules could react with the amino groups on CS chains, which transformed primary -NH2 groups into secondary -NH- groups. The peak of primary -NH2 groups was so close to secondary -NH- groups that there were not obvious changes between two curves. Moreover, the carboxyl groups on GO sheets could react with the amino groups, and the peak of formed secondary -NH- groups was also not obvious in FTIR spectra [25]. In the dissolution test of prepared nanofibers, CS/PVA nanofiber was dissolved in the 90 °C acidic aqueous solution (pH value 3-4), but CS/PVA nanofiber with 2.5 wt % GO not. It could indicate that the crosslinking actions occurred between amino groups on CS chains and epoxy, carboxyl groups on GO sheets in nanofibers.

[38]Yang, S.; Liu, Y.X.; Jiang, Z.X.; Gu, J.Y.; Zhang, D.W. Thermal and mechanical performance of electrospun chitosan/poly(vinyl alcohol) nanofibers with graphene oxide. Adv. Compos. Hybrid. Mater. 2018, 1, 722-730.

Reviewer 2 Report

The work entitled "Electrospun Chitosan/Poly (Vinyl Alcohol)/Graphene Oxide Nanofibrous Membrane with Ciprofloxacin Antibiotic Drug for Wound Dressing Application" is an interesting and compelling study. However there are some details that must be fixed before publication.

The entire work must be revised in terms of English writing. The entire manuscript is very poorly written, with wrong verbal conjugations and many grammar mistakes. This does not meet the standards of this journal. 

The introduction is clear and pertinent.

The materials and methods are well described with the exception of the "drug release assay", more detail should be given as it is very hard to understand how this process can be properly controlled and provide this kind of information. 

The authors refer in the abstract that the nanofibers are defect-free; however, if we look closely at images of Fig1, we see many beads and the lack of uniformity within the fibers. Also, there are many non-continuous fibers present in all images. The authors should have optimize the processing conditions of their fibers prior to the immobilization or functionalization with their drug. They should provide better images for this Figure as it decreases the quality of the entire work.

Quantitative antimicrobial testing should have been made together with this qualitative test. Even though the authors measured the diameter of the halo, there is always a certain degree of imprecision associated with it. Please provide an explanation for this decision. Also, there are no units in the table of Figure 8. 

Author Response

The work entitled "Electrospun Chitosan/Poly (Vinyl Alcohol)/Graphene Oxide Nanofibrous Membrane with Ciprofloxacin Antibiotic Drug for Wound Dressing Application" is an interesting and compelling study. However there are some details that must be fixed before publication.

The entire work must be revised in terms of English writing. The entire manuscript is very poorly written, with wrong verbal conjugations and many grammar mistakes. This does not meet the standards of this journal.

The introduction is clear and pertinent.

The materials and methods are well described with the exception of the "drug release assay", more detail should be given as it is very hard to understand how this process can be properly controlled and provide this kind of information.

The authors refer in the abstract that the nanofibers are defect-free; however, if we look closely at images of Fig1, we see many beads and the lack of uniformity within the fibers. Also, there are many non-continuous fibers present in all images. The authors should have optimize the processing conditions of their fibers prior to the immobilization or functionalization with their drug. They should provide better images for this Figure as it decreases the quality of the entire work.

Quantitative antimicrobial testing should have been made together with this qualitative test. Even though the authors measured the diameter of the halo, there is always a certain degree of imprecision associated with it. Please provide an explanation for this decision. Also, there are no units in the table of Figure 8.

Q 1) The entire work must be revised in terms of English writing. The entire manuscript is very poorly written, with wrong verbal conjugations and many grammar mistakes.

Thanks for your comment!

Answer: The entire manuscript has been revised in terms of English writing, and the revisions were marked in blue.

Q 2) The materials and methods are well described with the exception of the "drug release assay", more detail should be given as it is very hard to understand how this process can be properly controlled and provide this kind of information.

Thanks for your comment!

Answer: The “drug release assay” has been revised as followed:

The release characteristic in vitro of Cip and CipHcl from prepared nanofibrous membranes was characterized at 37 °C in release medium (PBS, pH of 7.2-7.4). The known mass (8 mg) of nanofibrous membranes were immersed in 15 mL PBS with stirring constantly. At designated time intervals, 2.0 mL solution was taken out, and an equal amount of fresh buffer solution was supplemented, assuring the amount of release medium constant during the release process. The amount of Cip and CipHcl which was released from nanofibrous membranes into PBS solution was quantified by capturing UV-Vis absorption spectra (TU-1901 Persee) at a wavelength of 280 nm. The release experiments of each sample were performed in triplicate, and the mean values were reported.

Q 3) The authors refer in the abstract that the nanofibers are defect-free; however, if we look closely at images of Fig1, we see many beads and the lack of uniformity within the fibers. Also, there are many non-continuous fibers present in all images. The authors should have optimize the processing conditions of their fibers prior to the immobilization or functionalization with their drug. They should provide better images for this figure as it decreases the quality of the entire work.

Thanks for your comment!

Answer: The processing conditions had been optimized to obtain the smooth nanofibers during the preparation process of nanofibers. Beads and non-continuous fibers are almost inevitable in the electrospinning nanofibers. When GO and antibiotic drugs were added into precursor solution, electrospinning process became more difficult. We have provided better images (Figure 1a) as follows according to your comment. The revised Figure 1 is as follows.

Figure 1. SEM images of drug-loaded nanofibrous membranes: (a) CS/PVA nanofibrous membrane, (b) CS/PVA/GO nanofibrous membrane, (c) CS/PVA/CipHcl nanofibrous membrane, (d) CS/PVA/GO/CipHcl nanofibrous membrane, (e) CS/PVA/Cip nanofibrous membrane, (f) CS/PVA/GO/Cip nanofibrous membrane.

Q 4) Quantitative antimicrobial testing should have been made together with this qualitative test. Even though the authors measured the diameter of the halo, there is always a certain degree of imprecision associated with it. Please provide an explanation for this decision. Also, there are no units in the table of Figure 8.

Thanks for your comment!

Answer: There are two main methods to investigate the antibacterial activity of materials, qualitative inhibition zone test and quantitative minimum inhibitory concentration test. In this paper, inhibition zone test was adopted due to that it was more intuitive to observe the antibacterial effect. At the same time, in order to improve the accuracy of experiment as far as possible, each group of inhibition zone test were repeated in triplicate, and then the average value was adopted. The results of the inhibition zone test are shown in Figure 8. In the future, more attention will be paid on the study of antibacterial activity from both qualitative and quantitative aspects.

Figure 8. Photographic images and radius statistical diagram of inhibition zones of drug-loaded nanofibrous membranes.

Reviewer 3 Report

  This paper prepared nanofibrous membranes based on chitosan, poly(vinyl alcohol) and  and graphene oxide (GO) for delivery of antibiotic drugs via electrospinning technique and investigated the effects of the incorporation of graphene oxide on the morphology, swelling, hydrophilicity, in vitro release behavior, and antibacterial acitivity of nanofibrous membranes.  This is an interesting contribution.  Comments and questions about details of data interpretation are given as follows:

1. Page 2, Line 76: Please provide the molecular weight of chitosan.

2. Page 3: What was the feed rate of the solution used during electrospinning?

3. The major composition of nanofibrous membranes is PVA, which is easily dissolved in water.  Thus, weight loss tests should be performed.  Moreover, the morphology of nanofibrous membranes after soaking in PBS should be given.

4. Page 4, Lines 175-176: “In the electrospun process, the addition of water-soluble CipHcl increased the viscosity of electrospun precursor solutions.” Please provide the viscosity data.

5. Figures 7 and 8: The error bar should be added. Moreover, the statistical analysis should be performed with these data.

6. Pages 10 and 11: The authors should explain the reason why the antibacterial activity CS/PVA/CipHCl (4.5 cm) against E. coil was lower than that of CS/PVA (9.77 cm).

Author Response

This paper prepared nanofibrous membranes based on chitosan, poly(vinyl alcohol) and graphene oxide (GO) for delivery of antibiotic drugs via electrospinning technique and investigated the effects of the incorporation of graphene oxide on the morphology, swelling, hydrophilicity, in vitro release behavior, and antibacterial acitivity of nanofibrous membranes.  This is an interesting contribution.

Comments and questions about details of data interpretation are given as follows:

Page 2, Line 76: Please provide the molecular weight of chitosan. Page 3: What was the feed rate of the solution used during electrospinning? The major composition of nanofibrous membranes is PVA, which is easily dissolved in water. Thus, weight loss tests should be performed. Moreover, the morphology of nanofibrous membranes after soaking in PBS should be given. Page 4, Lines 175-176: “In the electrospun process, the addition of water-soluble CipHcl increased the viscosity of electrospun precursor solutions.” Please provide the viscosity data. Figures 7 and 8: The error bar should be added. Moreover, the statistical analysis should be performed with these data. Pages 10 and 11: The authors should explain the reason why the antibacterial activity CS/PVA/CipHCl (4.5 cm) against E. coil was lower than that of CS/PVA (9.77 cm).

Q 1) Page 2, Line 76: Please provide the molecular weight of chitosan.

Thanks for your comment!

Answer: The average molar mass of chitosan was about 1.6×105 (measured by Ubbelohde viscometer), and the revision was as follows:

CS (deacetylation degree 90%-91%, average molar mass about 1.6 × 105).

Q 2) Page 3: What was the feed rate of the solution used during electrospinning?

Thanks for your comment!

Answer: In the experiment, the needle, displayed with a horizontal plane of 15°, was perpendicular to the collecting plate. And the electrospinning process could be carried out by the pushing from gravity of the electrospun precursor solution. We did not give the additional feed rate of the solution, and the moving of solution was pushing from gravity of the solution. It was evaluated that the feed rate was about 10 mL/6-8 h

Q 3) Page 3: The major composition of nanofibrous membranes is PVA, which is easily dissolved in water. Thus, weight loss tests should be performed. Moreover, the morphology of nanofibrous membranes after soaking in PBS should be given.

Thanks for your comment!

Answer: The residual weight of nanofibers (Figure 1) and the morphology of nanofibrous membranes after soaking in PBS (Figure 2) were given as followed. The weight of nanofibers decreased with the prolongation of immersion time. After 48 hours of immersion, the residual weight of GO loaded nanofibers membrane was higher. After immersion in PBS buffer, the nanofibers were broken by swelling, and many small fragments were formed in the solution.

Figure 1 Residual weight of the prepared nanofibers after soaking in PBS for designated time (6h, 12h, 24h, and 48h): (a) CS/PVA nanofibrous membrane, (b) CS/PVA/GO nanofibrous membrane, (c) CS/PVA/CipHcl nanofibrous membrane, (d) CS/PVA/GO/CipHcl nanofibrous membrane, (e) CS/PVA/Cip nanofibrous membrane, (f) CS/PVA/GO/Cip nanofibrous membrane.

Figure 2 The morphology of nanofibrous membranes after soaking in PBS for 48h: (a) CS/PVA nanofibrous membrane, (b) CS/PVA/GO nanofibrous membrane, (c) CS/PVA/CipHcl nanofibrous membrane, (d) CS/PVA/GO/CipHcl nanofibrous membrane, (e) CS/PVA/Cip nanofibrous membrane, (f) CS/PVA/GO/Cip nanofibrous membrane.

Q 4) Page 4, Lines 175-176: “In the electrospun process, the addition of water-soluble CipHcl increased the viscosity of electrospun precursor solutions.” Please provide the viscosity data.

Thanks for your comment!

Answer: The viscosity of electrospun precursor solutions was given as follows. As the addition of GO and antibiotic drugs, the viscosity of solutions increased gradually.

Figure 3 Kinematic viscosity (i) and Specific viscosity (ii) of nanofibrous membranes after soaking in PBS for 48h: (a) CS/PVA nanofibrous membrane, (b) CS/PVA/GO nanofibrous membrane, (c) CS/PVA/CipHcl nanofibrous membrane, (d) CS/PVA/GO/CipHcl nanofibrous membrane, (e) CS/PVA/Cip nanofibrous membrane, (f) CS/PVA/GO/Cip nanofibrous membrane.

Q 5) Figures 7 and 8: The error bar should be added. Moreover, the statistical analysis should be performed with these data.

Thanks for your comment!

Answer: According to reviewer's comment, the Figure 7 was revised. The modified Figure 7 is shown below. In Figure 8, the error bars were added and the statistical analysis was performed.

Figure 7. In vitro drug release curves of drug-loaded nanofibrous membranes.

Figure 8. Photographic images and radius statistical diagram of inhibition zones of drug-loaded nanofibrous membranes.

Q 6) Pages 10 and 11: The authors should explain the reason why the antibacterial activity CS/PVA/CipHCl (4.5 cm) against E. coil was lower than that of CS/PVA (9.77 cm).

Thanks for your comment!

Answer: The antibacterial activity of CS/PVA/CipHCl against E. coil was great, as reflected by the inhibition zone of 4.5 cm. The antibacterial activity of CS/PVA against E. coil was great, too. The inhibition zone test was referred by the previous papers (J. Agric. Food Chem. 2013, 61, 3901-3908). Inevitably, there is always a certain degree of imprecision associated with it. Thus we took several inhibition zone tests, and the preferable results and overall conclusions were obtained. In this inhibition zones test, the radius of inhibition zones of CS/PVA/CipHCl nanofibers was lower than CS/PVA nanofibers. But in general, the antibacterial activity of CS/PVA/CipHCl nanofibers was better than CS/PVA nanofibers.

Reviewer 4 Report

The manuscript describes preparation of electrospun CS/PVA/GO nanofibrous membranes for wound dressing applications, and presents their enhanced antibacterial activity by the addition of ciprofloxacin and ciprofloxacin hydrochloride.

The study is interesting but too preliminary at this stage for wound healing applications. The important outcome concerning cytotoxicity is not convincing. The source of melanoma cells is not provided. Melanoma cells are cancer cells and their viability potential is higher than that of primary melanocytes. Besides, melanocytes are not the major cells concerned by the wound healing process. Using primary fibroblasts and keratinocytes would be much more relevant in this study.

What would be the antibiotic realease in a wound environment or simply in the context of a cell culture ?

Minor comments

Information given in figure 1 is too small and can not be seen by the reader.

Author Response

The manuscript describes preparation of electrospun CS/PVA/GO nanofibrous membranes for wound dressing applications, and presents their enhanced antibacterial activity by the addition of ciprofloxacin and ciprofloxacin hydrochloride.

The study is interesting but too preliminary at this stage for wound healing applications. The important outcome concerning cytotoxicity is not convincing. The source of melanoma cells is not provided. Melanoma cells are cancer cells and their viability potential is higher than that of primary melanocytes. Besides, melanocytes are not the major cells concerned by the wound healing process. Using primary fibroblasts and keratinocytes would be much more relevant in this study.

What would be the antibiotic realease in a wound environment or simply in the context of a cell culture ?

Minor comments

Information given in figure 1 is too small and can not be seen by the reader.

Q 1) The study is interesting but too preliminary at this stage for wound healing applications. The important outcome concerning cytotoxicity is not convincing. The source of melanoma cells is not provided. Melanoma cells are cancer cells and their viability potential is higher than that of primary melanocytes. Besides, melanocytes are not the major cells concerned by the wound healing process. Using primary fibroblasts and keratinocytes would be much more relevant in this study.

Thanks for your comment!

Answer: The comment of reviewer was reasonable, primary fibroblasts and keratinocytes could be used in the experiments. We mainly considered the cytotoxicity of prepared nanofibers, Melanoma cells are a kind of human cells. Once nanofibers have great compatibility to melanoma cells, their cytocompatibility is relatively good.

Q 2) What would be the antibiotic realease in a wound environment or simply in the context of a cell culture?

Thanks for your comment!

Answer: The antibiotic released in a wound environment or simply in the context of a cell culture was different when the component of nanofibers was different. The antibiotic released from CS/PVA or CS/PVA/GO naofibers was CS. When antibiotic drugs were loaded into nanofibers, the antibiotic released from nanofibers mainly was antibiotic drugs. Meanwhile, little CS could be released to the environment. As a result, the nanofibers loaded with antibiotic drugs possessed the better antibacterial activity.

Q 3) Information given in figure 1 is too small and can not be seen by the reader.

Thanks for your comment!

Answer: The fiber diameter statistical charts were revised to be more obviously, and the revised figure is as follows:

Figure 1. SEM images of drug-loaded nanofibrous membranes: (a) CS/PVA nanofibrous membrane, (b) CS/PVA/GO nanofibrous membrane, (c) CS/PVA/CipHcl nanofibrous membrane, (d) CS/PVA/GO/CipHcl nanofibrous membrane, (e) CS/PVA/Cip nanofibrous membrane, (f) CS/PVA/GO/Cip nanofibrous membrane.

Round 2

Reviewer 1 Report

The supplied authors my critical remarks reply is mostly satisfactory and give some more clearance to the research concept. I would friendly recommend authors to be much self-critical in their future research and consider deeper insights in the results interpretation and experimental implementation as one of the major negative tendency in "modern science" is for its core fragmentation, masked for ex. by more and more "data analysis" shadow among others..  

Author Response

Thanks for your comment!

Answer: In our future research, we will be more self-critical, and consider deeper and further insights in the results. We will deliberate the investigation from the designing of experiments to analysis of results.

Reviewer 2 Report

The authors replied properly to all comments. Upon a second reading I still found some English mistakes that the authors did not noticed. Perhaps the journal edition process can assist with this. Overall this manuscript has merit and should be published.

Author Response

Title: Electrospun Chitosan/Poly (Vinyl Alcohol)/Graphene Oxide Nanofibrous Membrane with Ciprofloxacin Antibiotic Drug for Potential Wound Dressing Application. Line 23: …indicative to the great potential applications in wound dressing. Line 35: …considered as the excellent wound dressing substrates… Line 39: Nanofibrous membranes can also create a moist environment… Line 40: …area to promote a wound healing. Line 48: …moisture retention and readily available properties… Line 51: Poly (vinyl alcohol) (PVA) has been commonly used to enhance the electrospinnability of CS. Line 53: …biomechanical characteristics. Line 65: …for potential wound dressing applications, Line 66: …nanofibrous membranes was obtained by the addition of Ciprofloxacin (Cip) and Ciprofloxacin hydrochloride (CipHcl). Line 68: …via a scanning electron microscopy… Line 70: …via a fluorescence microscope. Line 71: …by a contact angle measurement. Line 84: , 0.01 molL-1 Line 148: …membranes was immersed in 15 mL PBS. Line 173: …were optimized during the process of the adjustment… Line 192: , which was relative wide… Line 194: …was concerned with the bend vibration of C-H bonds on PVA chains. Line 215: …relative regular crystal structures. Line 221: …the original crystal structures of… Line 223: Besides, the adsorption of drugs by GO was adverse to the gathering and formation of crystalline structure of drug molecules. Line 225: …destroyed the crystal structures… Line 237: , the membrane showed the… Line 254: …membranes could be conducive… Line 259: …from 299% to 270%. Line 283: …water drop and membrane Line 292: …that CS/PVA/Cip nanofibrous membrane and CS/PVA/CipHcl nanofibrous membrane showed the similar drug release profile trend. Line 297: , the release ratios of 91.1% and 59.0%... Line 299: …drug release ratios could be… Line 300: , promoting the release of Cip. Line 304: , the drug release profile trend of… Line 309: , the release ratios of 96.5% and 62.1%... Line 314: the drugs absorbed in GO nanosheets desorbed and released into the environment. Line 316: , the drug release profile trend of… Line 318: , promoting the release… Line 326: …investigated through inhibition zone tests, Line 326: …the results are shown in Figure 8. Line 328: , whose radius of the inhibition zone was 9.77 mm. Line 330: …. with the epoxy and carboxyl groups on GO nanosheets. Line 338: , promoting the enhanced antibacterial activity. Line 356: , promoting the applications in the field of wound dressings. Line 363: , after culturing without… Line 368: …was slightly lower, Line 380: …membranes loaded with CipHCl... Line 381: The structures of nanofibrous... Line 385: , nanofibrous membranes did not show... Line 388: In contact angle test, nanofibrous membranes showed the good After the addition of CipHcl and Cip, the contact angle increased slightly. Line 390: …regulated the drug.. Line 392: , promoting the.. Line 394: …enhanced the antibacterial activity of nanofibrous membranes, and the nanofibrous membranes were observed to achieve desirable antibacterial activity against... Line 396: …the samples was over 110%, Line 397: …the viability of MCs, Line 398: The entire results suggested that the prepared drug-loaded CS/PVA/GO nanofibrous membranes were great potential candidates as wound dressings.

Reviewer 4 Report

Since the cell issue has not been adequately addressed, and because the membranes are not suitable yet for wound healing applications, the reviewer thinks that the title should be changed accordingly.

Author Response

Thanks for your comment!

Answer: The comment of reviewer was reasonable, it was not suitable that the title was described for “Electrospun Chitosan/Poly (Vinyl Alcohol)/Graphene Oxide Nanofibrous Membrane with Ciprofloxacin Antibiotic Drug for Wound Dressing Application”. According to your comment, we would revised the title into “Electrospun Chitosan/Poly (Vinyl Alcohol)/Graphene Oxide Nanofibrous Membrane with Ciprofloxacin Antibiotic Drug for Potential Wound Dressing Application”.